# New-Onset Rheumatic Immune-Mediated Inflammatory Diseases Following SARS-CoV-2 Vaccinations until May 2023: A Systematic Review

**DOI:** 10.3390/vaccines11101571

**Published:** 2023-10-08

**Authors:** Arvind Nune, Victor Durkowski, S. Sujitha Pillay, Bhupen Barman, Helen Elwell, Kaustubh Bora, Syed Bilgrami, Sajid Mahmood, Nasarulla Babajan, Srinivasan Venkatachalam, Lesley Ottewell, Ciro Manzo

**Affiliations:** 1Department of Rheumatology, Southport and Ormskirk NHS Trust, Southport PR8 6PN, UK; 2Liverpool University Hospitals NHS Foundation Trust, Prescot Street, Liverpool L9 7AL, UK; victor.durkowski@nhs.net; 3Dartford and Gravesham NHS Trust, Dartford DA2 8DA, UK; ssujitha.pillay@nhs.net; 4Department of General Medicine, All India Institute of Medical Sciences (AIIMS), Guwahati 781101, India; bhupenbarman@aiimsguwahati.ac.in; 5BMA Library, BMA House, Tavistock Square, British Medical Association, London WC1H 9JP, UK; helwell@bma.org.uk; 6Haematology Division, ICMR-Regional Medical Research Centre, Dibrugarh 786001, India; kaustubhbora1@gmail.com; 7Department of Rheumatology, Royal Lancaster Infirmary, Lancaster LA1 4RP, UK; syed.bilgrami@mbht.nhs.uk (S.B.); lesley.ottewell@mbht.nhs.uk (L.O.); 8Department of Medicine, Southport and Ormskirk Hospital NHS Trust, Southport PR8 6PN, UK; miansajid.mahmood@nhs.net (S.M.); n.baba@nhs.net (N.B.); 9Department of Rheumatology, The Royal Wolverhampton NHS Trust, Wolverhampton WV10 0QP, UK; s.venkatachalam@nhs.net; 10Rheumatologic Outpatient Clinic, Azienda Sanitaria Locale Napoli 3, 80065 Sant’Agnello, Italy; manzoreumatologo@libero.it

**Keywords:** COVID-19 vaccines, rheumatic disease, immune-mediated inflammatory disease, vasculitis, connective tissue diseases, inflammatory arthritis, adverse events following immunization

## Abstract

A comprehensive, up-to-date systematic review (SR) of the new-onset rheumatic immune-mediated inflammatory diseases (R-IMIDs) following COVID-19 vaccinations is lacking. Therefore, we investigated the demographics, management, and prognosis of new R-IMIDs in adults following SARS-CoV-2 vaccinations. A systematic literature search of Medline, Embase, Google Scholar, LitCovid, and Cochrane was conducted. We included any English-language study that reported new-onset R-IMID in adults following the post-COVID-19 vaccination. A total of 271 cases were reported from 39 countries between January 2021 and May 2023. The mean age of patients was 56 (range 18–90), and most were females (170, 62.5%). Most (153, 56.5%) received the Pfizer BioNTech COVID-19 vaccine. Nearly 50% of patients developed R-IMID after the second dose of the vaccine. Vasculitis was the most prevalent clinical presentation (86, 31.7%), followed by connective tissue disease (66, 24.3%). The mean duration between the vaccine’s ‘trigger’ dose and R-IMID was 11 days. Most (220, 81.2%) received corticosteroids; however, 42% (115) received DMARDs such as methotrexate, cyclophosphamide, tocilizumab, anakinra, IV immunoglobulins, plasma exchange, or rituximab. Complete remission was achieved in 75 patients (27.7%), and 137 (50.6%) improved following the treatment. Two patients died due to myositis. This SR highlights that SARS-CoV-2 vaccines may trigger R-IMID; however, further epidemiology studies are required.

## 1. Introduction

The novel coronavirus disease of 2019 (COVID-19) is a highly contagious viral infection that results from Severe Acute Respiratory Syndrome Coronavirus 2 (SARS-CoV-2). The implications of this virus have been catastrophic, causing nearly seven million worldwide deaths as of July 2023 [1]. Subsequently, the COVID-19 pandemic prompted a global race to devise effective vaccines and curtail the spread of the virus. Developing a safe and effective vaccine requires several variables to be considered, including the type of vaccine, whether a carrier or vector would be used, adjuvant, excipients, dosage form, and the route of administration. Collectively, these variables can directly influence the immune responses induced and the resultant efficacy [2].

The most commonly used vaccines against COVID-19 include mRNA vaccines (specifically Pfizer-BioNTech and Moderna) and adenovirus vector vaccines (specifically Johnson and Johnson, AstraZeneca, Sputnik-V, and CanSino). Other vaccines include inactivated whole-virus SARS-CoV-2, such as Covaxin, Sinopharm, and Sinovac [3]. Nevertheless, despite their global rollout, data on vaccine responses regarding rheumatic immune-mediated inflammatory diseases (R-IMIDs) is scarce. Additionally, although COVID-19 vaccinations have been well received, there are several reports of new-onset R-IMIDs following COVID-19 vaccinations. Hence, the incidence of R-IMIDs, treatments, and prognosis of these patients must be investigated. This systematic review (SR) aims to critically review and summarise the clinical characteristics and patient demographics of new-onset R-IMIDs developing in adults following SARS-CoV-2 vaccinations.

## 2. Methodology

This SR was conducted according to the Preferred Reporting Items for Systematic Reviews and Meta-Analyses (PRISMA) guidelines [4]. The primary outcome of this review was to describe the demographics, clinical characteristics, treatment, outcomes, and timing of new-onset R-IMIDs following SARS-CoV-2 vaccinations.

### 2.1. Search Strategy

We conducted a systematic literature search on Medline, Embase, Google Scholar, LitCovid, and the Cochrane Library databases published between the 1st of January 2021 and the 30th of May 2023. The search strategy and PRISMA check list are available as Appendix A. We used indexing terms and relevant keywords to include all relevant studies. Three authors independently screened the article titles based on the inclusion criteria. Two authors reviewed the entire text of all the articles included. Disagreements (if any) were resolved through discussion and consensus with the senior author.

### 2.2. Inclusion and Exclusion Criteria

Literature was considered for inclusion in this review if it was: (a) published in the English language; (b) any adult case report or series, observational study, or randomised controlled trial that reported new-onset cases of R-IMID post-COVID-19 vaccination. Studies were excluded if the patient had existing flares of rheumatic diseases or if the research was conducted on animal models. Abstracts submitted at conferences or from non-peer-reviewed sources were not included.

### 2.3. Data Extraction

The following details were retrieved from the eligible studies: the country of publication, patient demographics, particulars about SARS-CoV-2 vaccination, new R-IMIDs onset time, clinical presentations, treatment, and disease outcomes. We used Microsoft Excel for the data extraction and management. 

### 2.4. Statistical Analysis

The included studies’ results were summarised in narrative form. Tabulated information was used to summarise the descriptive data on patient characteristics available from all studies. For continuous variables, the data were summarised by mean (standard deviation), median (range), and frequency/percentage, while for categorical variables, the mean (standard deviation) or median (range) were utilized. Continuous variables were expressed as the mean and standard deviation (SD). In this review, we did not use intervention or comparative descriptors. 

## 3. Results

### 3.1. Identification of the Literature

The search strategy returned 190 publications that contained information pertaining to 271 individuals [5,6,7,8,9,10,11,12,13,14,15,16,17,18,19,20,21,22,23,24,25,26,27,28,29,30,31,32,33,34,35,36,37,38,39,40,41,42,43,44,45,46,47,48,49,50,51,52,53,54,55,56,57,58,59,60,61,62,63,64,65,66,67,68,69,70,71,72,73,74,75,76,77,78,79,80,81,82,83,84,85,86,87,88,89,90,91,92,93,94,95,96,97,98,99,100,101,102,103,104,105,106,107,108,109,110,111,112,113,114,115,116,117,118,119,120,121,122,123,124,125,126,127,128,129,130,131,132,133,134,135,136,137,138,139,140,141,142,143,144,145,146,147,148,149,150,151,152,153,154,155,156,157,158,159,160,161,162,163,164,165,166,167,168,169,170,171,172,173,174,175,176,177,178,179,180,181,182,183,184,185,186,187,188,189,190,191,192,193,194]. All these publications were either case series or case reports. Medline, Embase, LitCovid, Google Scholar, and the Cochrane Library were searched. The details of the search strategy are summarised using the flow diagram in Figure 1, adapted from PRISMA guidelines.

### 3.2. Patient Demographics

The summary findings, including the country of origin, R-IMID diagnosis, and clinical outcome, are included in Table 1. The mean age of patients who developed new-onset R-IMID post-COVID-19 vaccination was 56 years (SD 20.2). Most of the patients were female (170, 62.5%). Most R-IMID cases were reported from the United States of America (47, 17.3%), Japan (36, 13.3%), followed by Italy (23, 8.5%), and Belgium (18, 6.6%). A history of various chronic diseases prior to vaccination was present in more than one-quarter of the patients. This included hypertension (29, 10.7%), autoimmune conditions (20, 7.4%), and heart disease (10, 3.7%). Table 2 represents the demographics of the patients included in this review.

### 3.3. Vaccination Characteristics

The majority of the patients developing R-IMID had received the Pfizer BioNTech vaccine (153, 56.5%), followed by the Oxford-AstraZeneca (61, 22.5%) and Moderna (33, 12.2%) vaccines. Most patients (123, 45.4%) had received at least two doses at the time of new R-IMID onset. Table 3 represents the data on SARS-CoV-2 vaccinations before patients developed R-IMIDs. A total of 11 patients developed new-onset R-IMID after receiving a booster dose of the vaccine.

### 3.4. Clinical Presentations

Vasculitis was the most common clinical presentation (86, 31.7%), followed by connective tissue diseases (66, 24.3%) and inflammatory arthritis (55, 20.3%). Table 4 depicts the distribution of the new-onset R-IMID following SARS-CoV-2 vaccinations. Small vessel vasculitis was the most common (64,76%) vasculitis, and antineutrophilic cytoplasmic antibody (ANCA)-associated vasculitis (33, 51%) was the most reported form of small vessel vasculitis. Among the connective tissue diseases (CTD), idiopathic inflammatory myositis (IIM) (37, 56%) was reported most often. The new R-IMID onset following administration of the ‘trigger’ dose of the COVID-19 vaccine ranged from 1–90 days, with a mean of 11.0 days. Corticosteroids were administered in most patients (220, 81.2%). Over 40% (115) of patients were treated with steroid-sparing drugs such as methotrexate (25, 9.2%), hydroxychloroquine (HCQ) (26, 9.5%), cyclophosphamide (23, 8.4%), mycophenolate (15, 5.5%), IV immunoglobulin (IVIG) (12, 4.4%), plasma exchange (10, 3.6%), anakinra (7, 2.5%), tocilizumab (8, 2.9%), or rituximab (18, 6.6%). Complete remission was achieved in 75 patients (27.7%), while a marked improvement in all symptoms was observed in 137 patients (50.6%). Eight patients were admitted to the intensive care unit (ICU) to manage their disease. Two deaths were reported because of myositis and rhabdomyolysis.

We have summarised the key characteristics of the most common new-onset R-IMIDs under various sections.

### 3.5. Vasculitides

#### 3.5.1. Small-vessel Vasculitis

The three most common small vessel vasculitis were ANCA-associated vasculitis (AAV) (33/64, 51%), cutaneous vasculitis (22/64, 34%), and Henoch-Schoenlein purpura (HSP) (5/64, 7%). Out of AAV patients, the majority (20) had positive Myeloperoxidase Antibody (MPO) ANCA with titres ranging from 1.1 units/mL to 3500 units/mL [41,42,43,44,45,46,48,96,116,122,124,128,137,150,160,168,178,182,183,187], and nine were proteinase-3 (PR3) ANCA positive, ranging from 6.7 units to 259 units. Of AAV, 24/30 patients developed biopsy-proven crescentic glomerulonephritis [38,41,42,45,46,48,51,52,55,96,116,122,124,128,130,136,137,150,157,160,168,178,182,187]. Six had pulmonary and renal syndrome [45,55,137,157,160,178], whereas three patients had only lung involvement with pulmonary haemorrhage or multi-focal consolidations [43,109]. Two patients were positive for both PR3 and MPO-ANCA [43,44]. A total of seven patients received plasma exchange: three patients for pulmonary haemorrhage and the remaining four for renal vasculitis [43,48,55,130,150,160,178]. Only three patients were presented with ANCA-negative vasculitis, and their symptoms improved with treatment [37,55,148]. Out of this, one patient developed ANCA-negative vasculitis following the Pfizer-BioNTech vaccine. The patient had enhancing nodules in bilateral lungs on chest computed tomography (CT) and crescentic glomerulonephritis on kidney biopsy (49). He was successfully treated with IV methylprednisolone. Another patient had crescentic vasculitis following haematuria and acute kidney injury (AKI). The simultaneous occurrence of systemic symptoms and AKI soon after the second dose of the Moderna vaccine supports a causative relationship. IV corticosteroids and cyclophosphamide were used to treat the patient successfully [55]. The third patient, a 77-year-old male, had granulomatous vasculitis on the kidney biopsy after the AstraZeneca SARS-CoV-2 vaccine (37). One patient had cryoglobulinaemic vasculitis [167].

#### 3.5.2. Medium-Vessel Vasculitis

A total of six patients belonged to medium vessel vasculitis (MVV) [36,49,54,104,111,141]. Among the MVV patients, one had cutaneous polyarteritis nodosa (PAN) [54], and the remaining five had systemic PAN. Out of these, one patient presented with acute anterior uveitis, and magnetic resonance angiography (MRA) of the abdomen was suggestive of vasculitis involving the celiac trunk [49], and another patient had associated myalgia, and a muscle biopsy of the left gastrocnemius also showed fibrinoid necrosis [111]. Another patient had an acute kidney injury with a kidney biopsy that showed features of necrotising vasculitis and required cyclophosphamide and haemodialysis for 32 days [36]. 

#### 3.5.3. Large-Vessel Vasculitis

Of eight patients, four had IgG4 disease [184,185], and four had large vessel vasculitis (LVV) [39,40,117,166]. One patient had bilateral submandibular gland swelling and raised C-reactive protein (CRP) and IgG4 at 1100 mg/dL (11–121). A positron emission tomography/computed tomography (PET-CT) showed fluorodeoxyglucose (FDG) uptake in the pancreas [185]. The patient was treated with a tapering dose of prednisolone. Another patient had breathlessness, fever, and recurrent pleural effusion. Her IgG4 was raised. Thoracoscopy showed multi-loculated pleural effusions. Histopathology showed lymphocytic infiltrates and positive IgG4-positive plasma cells. The patient had thoracocentesis, which improved the symptoms. However, the patient subsequently failed to attend a follow-up review [184]. 

The rest of the four patients had PET-CT evidence of FDG uptake in large vessels following their presentation with fever (3/4) and neck and peripheral joint pains. All four patients had raised inflammatory markers (CRP and erythrocyte sedimentation rate (ESR)). Two patients received oral corticosteroids; one was treated with IV tocilizumab [117]. One patient had transient LVV, as the patient’s symptoms resolved entirely after taking naproxen for 2 weeks [166]. This patient’s PET-CT scan showed increased FDG uptake in bilateral brachial, subclavian, and carotid arteries. Treatment details were not provided for two patients [39,40]. 

#### 3.5.4. Giant-cell Arteritis

All patients were diagnosed with giant-cell arteritis (GCA) [31,32,33,34,114,131,153], except one who was less than 50 years old [35]. Two patients had normal CRP. Of eight patients with confirmed GCA, three had bilateral temporal arteritis (TA) [31,131,153]. One of them also had scalp necrosis in the parietal-temporal area [31]. Another patient had bilateral visual complaints and lost total vision in the left eye [131]. This 68-year-old man had ocular CT evidence of ischemic optic neuropathy in the left eye and retinal ischemic changes in the right eye. The patient lost vision despite IV methylprednisolone and tocilizumab. A 34-year-old man developed central retinal artery occlusion (CRAO) and central retinal vein occlusion (CRVO) and subsequently developed ischemic optic neuritis following blurred vision in the left eye after receiving the SARS-CoV2 vaccination [35]. His CRP was normal; however, his ESR was marginally raised at 26. Another patient with normal CRP had left temporal artery induration and confirmed left TA on biopsy [32]. One out of eight patients had no headache; however, PET-CT showed widespread FDG activity of the aortic branches, including cranial arteritis [33]. A temporal artery biopsy confirmed the diagnosis in five out of six patients [31,32,114,131,153]. The patient with a negative temporal artery biopsy had positive FDG activity on PET-CT. PET-CT confirmed LVV in 3/4 of patients [33,34,114]; however, one patient with negative PET-CT had a positive temporal artery biopsy [32]. All patients received corticosteroids except one, for whom the authors did not provide the treatment details. Two patients received tocilizumab [114,131], and one patient was treated with methotrexate [34].

### 3.6. Connective Tissue Diseases

#### 3.6.1. Idiopathic Inflammatory Myositis

Out of 24 patients with dermatomyositis (DM), all had skin rashes, with many having Gottron’s papules [11,56,58,59,62,65,66,103,112,132,147,149,152,162,190]. One-fifth of patients (6) had no muscle weakness [11,65,103,112,147]. A total of eight patients had interstitial lung disease (ILD) [59,65,103,112,152,190]. Of these, two patients also had myocarditis [190]. One of these patients, who had a positive anti-alanyl-tRNA-synthetase antibody (PL-12), had raised troponin-T [190]. One patient with a positive topoisomerase-1 (Pm-Scl-70) antibody had a confirmed DM on a muscle biopsy, dark urine, and superimposed rhabdomyolysis [62]. This 44-year-old male also had raised creatine kinase (CK) at 151,058 without any clinical features of systemic sclerosis. He developed compartment syndrome in multiple limbs and died despite being treated with IV methylprednisolone and cyclophosphamide. Only two (8.3%) patients with DM were found to have malignancies [66]. One had positive faecal occult blood and sigmoid colon cancer, while another had positive cancer antigen 19-9 (Ca-19-9), followed by CT scan evidence of colon cancer. Both patients had positive anti-transcription intermediary factor 1 (TIF-1) antibodies. The other three patients with TIF-1 antibodies had no evidence of malignancy on extensive cancer screening [65,132,147]. Twenty-two patients (91.6%) had a positive myositis antigen profile (MAP). Of the two patients with no MAP, one had a positive ribonucleoprotein (RNP) antibody [111]. The most common antibody was anti-melanoma differentiation-associated gene 5 (anti-MDA-5), followed by Ro-52 and anti-TIF-1. Ro-52 was co-presented with an anti-MDA-5 antibody in seven patients [11,65,112,152]. One patient with a positive Pm-Scl-70 antibody for DM had no clinical features of systemic sclerosis [62]. One patient developed anti-MDA-5 antibody-positive amyopathic DM and insulin-dependent diabetes mellitus simultaneously at 62 years following the Pfizer-Bio-N-Tech vaccination [103]. Although she had a normal CK, CT chest evidence of ILD and a positive skin biopsy confirmed DM. Despite the patient being treated with IV methylprednisolone, cyclophosphamide, and plasmapheresis, she died following mediastinal and subcutaneous emphysema. All patients received corticosteroids. Twenty-one (87.5%) patients received multiple immunosuppressants; eleven needed mycophenolate [11,56,59,65,112,132,152,190], nine needed IVIG [56,65,66,112,132,152,190], five needed cyclophosphamide [56,62,65,103], five needed rituximab [65,112,152], four patients each needed methotrexate [58,65,152], azathioprine [65,149,152] or tacrolimus [65,103], and two required plasma exchange [65,103]. Two patients were treated with tofacitinib [65,152], one with anakinra [152], and one with associated ILD was treated with nintedanib and daratumumab [152].

Among nine patients with polymyositis (PM) [56,57,61,102,125,144,154], three had fever [57] as a presenting symptom along with myalgia and muscle weakness; one had skin rashes [61]; all nine patients had muscle pain and weakness. CK was raised in four patients (range 7790–22,000) [56,102,125,144], normal in three patients [57,61,154], and not recorded in two patients. In three patients with myalgia, a muscle biopsy confirmed PM [61,125,144]; however, no muscle biopsy (MB) was performed in four patients [56,57,154], and it was negative in one patient [123]. The patient with negative MB had magnetic resonance imaging (MRI) and electromyography (EMG) evidence of myositis; however, the myositis antigen profile was not checked. Among the patients who had no MB, all had MRI scan evidence of myositis. Among these, two patients had normal CK [57], one not recorded [154], but one had raised CK at 10,222 [56]. Among the nine PM patients, only one had a malignancy (pancreatic cancer) [56]. One patient had confirmed PM on MB and co-existing sweet syndrome on a skin biopsy [61]. This 37-year-old man also had a positive Pm-ScL70 antibody but had no clinical features for systemic sclerosis. All patients received corticosteroids; six patients received steroid-sparing drugs such as IVIG (four patients) [56,61,102,144], mycophenolate (one patient) [57], and azathioprine (one patient) [144].

#### 3.6.2. Systemic Lupus Erythematosus

Out of eighteen patients with systemic lupus erythematosus (SLE), two developed SLE with renal involvement [105,131], the other two developed cardiac involvement [146,147], one with secondary antiphospholipid (APL) syndrome [146], and another one had acute pancreatitis [148]. A middle-aged male with known Sjogren’s syndrome developed skin rashes and pancytopenia for new-onset lupus, with raised Ds-DNA > 300. He had proteinuria and new confusion. The patient was diagnosed with renal and neuropsychiatric lupus. He was successfully treated with high-dose corticosteroids and mycophenolate. [105]. Another young female developed widespread, painful, tender skin nodules with positive ANA and Smith antibodies. A skin biopsy confirmed neutrophilic urticarial dermatosis. The patient responded well to oral corticosteroids and HCQ [124]. Another patient presented with skin rashes and autoimmune haemolytic anaemia following vaccination. She had positive ANA, a Coombs test, and thrombocytopaenia. The patient was diagnosed with SLE and Evans syndrome and was successfully treated with oral corticosteroids [128]. Another 23-year-old female with no medical history developed facial and limb oedema and anasarca a week after receiving the SARS-CoV-2 vaccination. She only had weakly positive Ds-DNA 14 U. However, she had 12 g of proteinuria. Her kidney biopsy 2 weeks after immunisation confirmed class V lupus nephritis. This was treated with high-dose oral corticosteroids and mycophenolate [131]. A middle-aged lady with known miscarriages presented a week after Pfizer-BioNTech with polyarthralgia, breathlessness, and raised D-dimer. A CT pulmonary angiogram confirmed an embolism. The patient subsequently developed plural and pericardial effusions, leading to cardiac tamponade. She tested positive for ANA, ds-DNA, cardiolipin, and beta-2 glycoprotein antibodies; therefore, she was diagnosed with SLE with secondary APL syndrome. Her clinical condition improved with the corticosteroids azathioprine, HCQ, and warfarin [146]. A young lady developed nausea and vomiting soon after the COVID-19 vaccination. She had positive ANA, ds-DNA, and bulky pancreas on a CT scan. She was diagnosed with SLE and acute pancreatitis. She responded well to IV methylprednisolone, azathioprine, and HCQ [148]. 

#### 3.6.3. Subacute Cutaneous Lupus

Of six patients with subacute cutaneous lupus (SACL), three developed widespread skin rashes on the trunk and limbs [117,141,142]. The remaining patients had localised cutaneous rashes, mainly on the face. Anti-Ro antibodies were positive in all six patients, and SACL was confirmed on skin biopsy [117,125,140,141,142,143]. A skin biopsy revealed hyperkeratosis in the epidermis and the dermal infiltration of lymphocytic and histiocytes. One patient was also positive for anti-histone and another for anti-ribosomal antibody [140,143]. Three patients were started on HCQ [117,140,141], and one patient with diffuse SACL was treated with mycophenolate and IVIG [141].

### 3.7. Inflammatory Arthritis

#### 3.7.1. Reactive Arthritis

Out of 30 reactive arthritis (ReA) patients, the majority presented with oligoarthritis (17,56%) [121], followed by monoarthritis (7, 23%) [9,29,69,97,98,107,135], and polyarthritis (6,20%) [86,98,158,169]. Among the patients with monoarthritis, four had knee joints affected [9,97,98,135], two had elbow joints affected [69,107], and one had ankle joints affected [29]. CRP was raised in 20 patients, between 10 and 237. However, eight patients had normal CRP [98,121], and no CRP was available for two patients [69,86]. A total of twenty-five patients received corticosteroids, two nonsteroidal anti-inflammatory drugs (NSAIDs) [98], and three patients received a combination of NSAIDs and corticosteroids [69,97,158]. Of the twenty-eight patients who received corticosteroids, most received oral corticosteroids (twenty-two patients) [29,86,98,121,169], four received intraarticular (IA) corticosteroids [9,67,97,107], one received intramuscular corticosteroid (IM) [158], and one received IV corticosteroids. 

#### 3.7.2. Rheumatoid Arthritis

Among the patients with rheumatoid arthritis (RA), twelve [8,11,101,113,115,139,140,191] were seronegative, and nine [69,101,129,133,164,191,192] were seropositive for anti-cyclic citrullinated peptide (CCP) antibodies or rheumatoid factor (RF) or both. Among these, six patients [69,101,129,164,193] had both RF and CCP positives, and three patients [101,133,191] were only positive for RF. The majority (seventeen patients, 80%) had polyarthritis presentation [8,11,101,113,115,129,164,191,192], whereas two had oligoarthritis [69,133] and another two had monoarthritis presentation [139,140]. A total of eighteen patients received corticosteroids, eleven patients received disease-modifying antirheumatic drugs (DMARDs), six received methotrexate [69,101,133,164,192], two received leflunomide [101,191], one received mycophenolate [11], three received HCQ [113,164,192], one received sulphasalazine [164], and four patients received biologic drugs: tocilizumab for one [133], etanercept for one [115], adalimumab for one [101], and golimumab for one patient [101]. Nine patients had corticosteroids as monotherapy [8,11,129,139,140,191], another ten received prednisolone and DMARD in combination, and one patient with intolerance to methotrexate and leflunomide received adalimumab [101]. Although many cases did not comment on whether their patients had erosions, one seropositive patient (RF 130 units, CCP 250 units) with polyarthritis following the Pfizer BioNTech vaccination had erosions in the hands and feet [101]. However, it was unclear how soon, following the RA diagnosis, the patient developed erosions. 

### 3.8. Polymyalgia Rheumatica

Out of 21 patients with confirmed polymyalgia rheumatica (PMR), all except one [60] had typical inflammatory shoulder and thigh girdle pain and stiffness. All patients had raised inflammatory markers, including ESR and CRP [6,11,27,28,29,30,94,101,120,123,146]. One patient had no shoulder symptoms but had increased FDG uptake of cervical, lumbar interspinous bursae, ischial tuberosities, trochanteric bursae, and hips, but no PET-CT PMR features in shoulders [60]. A total of 12 patients had PET-CT scans to evaluate PMR features, but none had malignancy [27,28,30,120]. No patients had confirmed TA, but two patients had jaw claudication symptoms along with PMR [27,94]. The authors did not mention further details about one of these patients, including whether the patient had any investigations for jaw claudication [94]. Another patient with jaw claudication had a PET-CT scan, which showed no increased FDG uptake on cranial vessels [27]. However, this patient had a typical PMR distribution of FDG uptake on PET-CT and a raised CRP. The patient’s symptoms improved on acetaminophen alone, and she was in remission after 5 weeks of review. All patients received oral corticosteroids except one [27]. The prednisolone dose varied between 15 and 25 mg daily, and a few patients received prednisolone dosing based on 0.3 mg per Kg of body weight. Steroid-sparing drugs were offered to four patients; three received methotrexate and one received tocilizumab [28]. 

### 3.9. Adult-Onset Stills Disease

All 22 patients with adult-onset stills disease (ASOD) had a fever as a presenting complaint [7,74,87,88,89,90,92,95,99,118,119,159,171,186]. Although characteristic salmon-pink skin rashes were also a common presenting feature along with arthralgia/arthritis, it was a delayed presenting feature in five patients [74,87,88,89,118]. Serum ferritin and CRP were raised in all patients. Three patients developed macrophage activation syndrome (MAS) [7,92,186], and three developed hemophagocytic lymphohistiocytosis (HLH)/hemophagocytic syndrome (HPS) [92,186]. Overall, six patients had cardiac involvement [7,74,88,92,95,118], and one had lung involvement [89]. All patients except one (naproxen) received corticosteroids [7]. While six patients only needed corticosteroids, fifteen (68%) patients required multiple DMARDs to treat their condition: six required anakinra [88,92,186], four required tocilizumab [7,74,186], and one required cyclophosphamide [92]. Four patients required methotrexate [7,95,119,171] and two required cyclosporin [186] as maintenance therapy. Among the patients with cardiac involvement, two had bilateral pleural and pericardial effusions [74,88]. Two other patients developed myocarditis and heart failure [95,118]. One patient without any prior cardiovascular disease developed acute heart failure with a left ventricular ejection fraction of 15–20%, which returned to normal after treatment with IV methylprednisolone and methotrexate therapy. Among the three patients with MAS, one with existing AOSD developed MAS [186]. The patient developed pancytopenia, raised D-dimer, and had a very high ferritin level of 136,000. The patient was successfully treated with methylprednisolone, IVIG, anakinra, and maintenance therapy with cyclosporin. The second patient developed MAS despite ASOD being treated with corticosteroids, IV cyclophosphamide, and subcutaneous (SC) anakinra. Switching to IV anakinra improved the platelet count, and the patient was in remission [92]. The third, with MAS, developed very high ferritin >100,000 and diffused alveolar haemorrhage and needed corticosteroids, IVIG, and tocilizumab [7]. Among the patients with ASOD-associated HLH/HPS, a 55-year-old female developed thrombocytopenia, hyperferritenemia of 38,101, hepatosplenomegaly, and deranged liver function tests (LFTs). A liver biopsy confirmed phagocytosis of erythrocytes [186]. Tocilizumab and corticosteroids improved the patient’s condition. Another 77-year-old female patient developed hyperferritenemia (a value of 48,377), thrombocytopenia, and anaemia. The bone marrow biopsy has confirmed HPS [186].

## 4. Discussion

This SR explored the development of new-onset R-IMIDs in adults after receiving SARS-CoV-2 vaccinations. To the best of our knowledge, this is the first SR to comprehensively investigate the emergence of new-onset R-IMIDs following SARS-CoV-2 vaccinations. Our findings indicate that, despite the substantial rollout of the COVID-19 vaccinations, few cases of new-onset R-IMIDs have been reported. Overall, 271 cases of various R-IMIDs have been reported in the literature. Most of the patients had received the Pfizer BioNTech vaccine. The three most common conditions reported in our SR following vaccination were vasculitis, CTDs (mainly IIM and SLE), and inflammatory arthritis. Corticosteroids (221, 81.5%) were used to treat the vast majority of patients. Only eight patients were admitted to the ICU for disease management, with the vast majority (212, 78%) experiencing disease remission or improvement following the treatment. Two patients died because of myositis and rhabdomyolysis. Two AAV patients with lung involvement survived following their ICU admission. Most cases were characterised by symptomatic improvement and complete remission after medical intervention. Another SR published in 2022 studied new-onset arthritis following SARS-CoV-2 vaccinations. Among the 45 patients they studied, the majority (64.4%) developed joint symptoms within the first week of vaccination, predominantly after the first dose [195].

Previously, a study reported 66 cases of short-term inflammatory musculoskeletal manifestations after administering the COVID-19 vaccination across 16 Italian rheumatology centres. Most patients in that report were found to have received the Pfizer BioNTech vaccine (59%), with the onset of arthritis occurring between 11 and 13 days after the ‘trigger’ vaccine dose. Their management approach mirrors our observations in this review, with glucocorticoids, non-steroidal anti-inflammatory drugs, and analgesics employed in most instances. DMARDs were considered on a case-to-case basis. Almost all patients (74%) with PMR-like onset achieved complete remission within two weeks, while 67% of those who developed polyarthritis still presented with active disease at the six-week follow-up [196]. An SR analysing 2184 patients with myocarditis following the SARS-CoV-2 vaccination revealed that most patients received the mRNA-based vaccine (99.4%). Similar to our study, their patients’ mean duration from vaccination to symptom onset was also short. The mean duration in our study between receiving the ‘trigger’ dose of the vaccine and developing R-IMID was 10.6 days, whereas the mean time for their patients to develop myocarditis symptoms was 4.01 ± 6.99 days. Similar to our cohort, most myocarditis patients responded well to conservative treatment such as non-steroidal anti-inflammatory drugs or corticosteroids with a good prognosis, although six patients died [197]. Another study from Iran documented 14 patients with different autoimmune rheumatic diseases (ARDs) as sequelae to COVID-19 vaccinations. Despite the relatively small numbers, ARDs were more frequent among those who received the AstraZeneca vaccines than in individuals who received other vaccines [198]. Both Pfizer BioNTech and Moderna vaccines are mRNA vaccines. Thus, there appears to be a link between the type of vaccine and the subsequent de novo emergence of R-IMIDs. Furthermore, the mRNA contained in vaccinations may cause autoimmunity by activating the inflammasome pathway, which is recognized by toll-like receptors [199]. 

Molecular mimicry is the leading theory put forth to account for the emergence of these autoimmune diseases. The antigen included in the vaccine, known as an adjuvant (aluminium salts, virosomes, oil-in-water emulsions, immune modulatory complexes, squalene, montanide, lipovant, and xenobiotic adjuvants), is thought to share structural similarities with self-antigens. The activation of “innocent bystanders”, which results in autoreactive T cells, polyclonal activation, and epitope dissemination, is another possibility; nevertheless, the pathogenic processes underlying the association between vaccinations and autoimmune disorders are not entirely understood [200]. 

Another study suggests that the occurrence of R-IMIDs in succession with anti-SARS-CoV-2 vaccination arises due to the anti-SARS-CoV-2 spike antibodies or the SARS-CoV-2 recognising T-cells that subsequently trigger prolonged immune-mediated inflammation [11]. It is possible that toll-like receptors (TLRs) 7 and 9 could be the common link between PMR (or PMR-like syndromes) and mRNA vaccines, which, in those with a genetic predisposition, influence the excessive production of inflammatory cytokines (such as IL-6) [30]. The manufacture of many cytokines necessary for the innate immune response is triggered by TLRs’ detection and signalling within endo-lysosomal compartments. For instance, PMR patients’ peripheral mononuclear blood cells exhibit elevated TLR7 and TLR9 expression, disappearing once PMR is in complete remission [201]. The substantial activation of TLR signalling following immunisation with the Pfizer BioNTech vaccine was recently shown by an observational investigation using transcriptional signatures in whole blood samples of healthy volunteers [202]. To our knowledge, TLR7 and TLR9 levels have not been measured in individuals who developed PMR or PMR-like disorders after receiving COVID-19 mRNA vaccinations.

Furthermore, it has been suggested that age-associated B cells (ABC) take part in the immunological response brought on by the SARS-CoV-2 vaccine. These ABC cells, also known as double negative or CD11c + T-bet + cells in humans, grow with age in healthy people and are more prevalent at an earlier stage in autoimmune disorders. Immunoglobulin G production, increased antigen presentation to T cells, and germinal centre development are characteristics of the cells. The ability of these ABC cells to elicit a hyper-response capable of producing autoreactive antibody-secreting plasma blasts in reaction to TLR-7 signalling is another feature of these cells. TLR-7/8 and TLR-9 agonists are used in mRNA/DNA SARS-CoV-2 vaccinations as “adjuvants,” which may encourage the subset of ABC to produce autoantibodies and post-vaccination autoimmune disorders [203,204].

Our SR found that the current literature on new-onset R-IMIDs following COVID-19 vaccinations is constituted exclusively by case series and case reports. Consequently, the evaluation of association or cause-effect evidence between COVID-19 vaccines (i.e., the suspected risk factor) and new-onset R-IMIDs (i.e., the potential outcome) was precluded as it is outside the scope and rigour of case series and case reports. 

However, we speculate whether these R-IMIDs qualify to be labelled as adverse events following immunisation (AEFI) following COVID-19 vaccines. According to the WHO recommendations [205], AEFI is defined as “any untoward, unfavourable, or unintended medical occurrence that occurs after immunisation and does not necessarily have a causal relationship with the use of the vaccine. This adverse event could be a symptom, disease, abnormal laboratory finding, or unintentional sign”. To determine the cause of an AEFI, the WHO recommendations suggest the following four steps: assessment of the temporal relationship between vaccination and AEFI; a plausible time duration between vaccination and AEFI; exclusion of other causes, such as medications the patient is taking or comorbid conditions; and appraisal of the causal relationship based on the existing literature.

The short time span between COVID-19 vaccine administration and the onset of R-IMIDs suggests the potential possibility of a cause-and-effect relationship. It is possible that the biological mechanisms linking the new-onset R-IMIDs to COVID-19 vaccination are perhaps similar to those of R-IMIDs arising from post-COVID-19 infection and that a genetically predisposed individual could develop an autoimmune condition when exposed to an environmental trigger. 

Finally, we observed that the patients with new-onset R-IMIDs following the administration of COVID-19 vaccines were reported from the developed as well as the developing world. The fact that these R-IMIDs were not restricted to a particular geographical region suggests the global relevance of this phenomenon. Most R-IMID cases have been reported from five of the six World Health Organisation (WHO) regions (namely Europe, Americas, Eastern Mediterranean, Western Pacific, South East Asia), with only four cases reported from the region of Africa. This conspicuously low number of reports from Africa hitherto in the R-IMID reporting landscape after COVID-19 vaccination may be due to the relatively low vaccination rates compared to other parts of the world.

### Strengths and Limitations

The strength of this study is that it is the first to conduct an up-to-date and comprehensive study of all reported cases from all subgroups of R-IMIDs following SARS-CoV-2 vaccines using five different databases. This study’s findings will help clinicians provide better patient care and, hopefully, pave the way for further research into establishing a cause-and-effect association. However, there were a few limitations to our SR. This study is primarily descriptive and consists of only case reports and series with no long-term follow-up data. Thus, the level of evidence is low, and there is a risk of reporting bias. Notably, this also illustrates the existing gaps in knowledge and the need to initiate formal epidemiological studies to address such gaps in this emerging area. 

## 5. Conclusions

Our review suggests that R-IMIDs may develop after administering COVID-19 vaccines to adults. The onset of symptoms after taking the COVID-19 vaccine is short, with many patients developing acute clinical symptoms with manifestations of R-IMIDs. Vasculitis was the most reported condition, followed by CTDs and inflammatory arthritis. However, the association of COVID-19 vaccines with R-IMID development has yet to be conclusively answered. Although many cases of R-IMIDs are being reported across different parts of the world, R-IMIDs following the COVID-19 vaccinations are still rare, short-lived, and respond to steroidal and other immunosuppressive agents, and therefore have a good prognosis.

## Figures and Tables

**Figure 1 vaccines-11-01571-f001:**
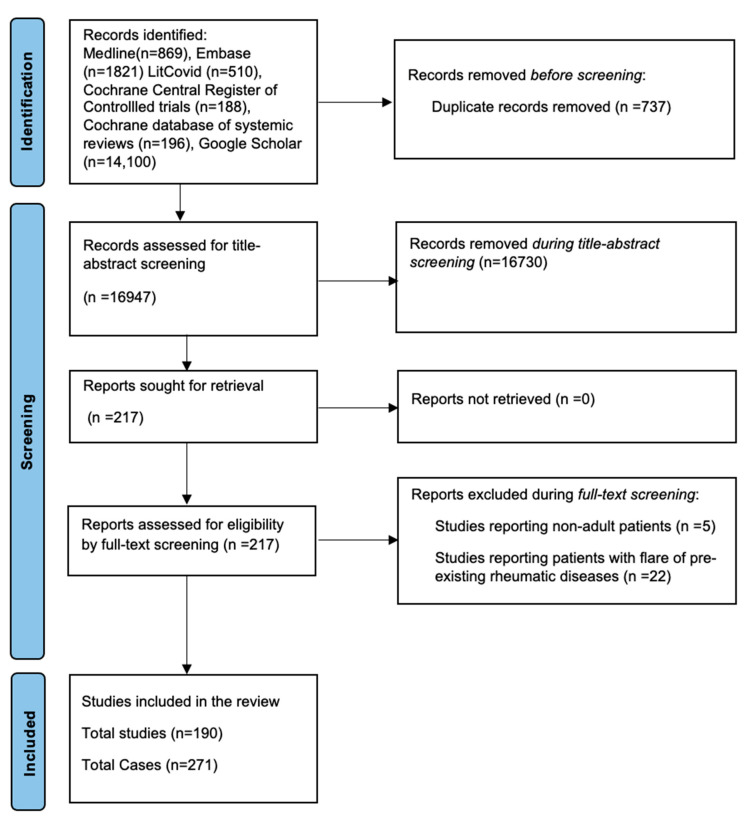
PRISMA flow diagram of data extraction of the studies included in the systematic review.

**Table 1 vaccines-11-01571-t001:** Demographics, diagnosis, and clinical outcomes of the new-onset R-IMIDs following the SARS-CoV-2 vaccinations.

SN	Article	Country	Age	Sex	Vaccine Received	R-IMID Diagnosis	Immunosuppressive Drugs Used	Clinical Outcome
1	Shimagami et al. [5]	Japan	90	F	PfizerBioNTech	Tenosynovitis and pleural effusion	Prednisolone	Clinical improvement
			70	M	PfizerBioNtech	Tenosynovitis	Prednisolone	Clinical improvement
2	Osada A et al. [6]	Japan	80	F	Pfizer BioNTech	PMR	Prednisolone	Clinical improvement
3	Padiyar et al. [7]	India	20	F	Oxford-AstraZeneca	AOSD	corticosteroids, naproxen, and tocilizumab	Clinical improvement
			47	M	Oxford-AstraZeneca	AOSD	Naproxen	Clinical improvement
			35	F	Oxford-AstraZeneca	AOSD	Steroid, tocilizumab	Clinical improvement
4	Unal Enginar et al. [8]	Tukey	74	F	Sinovac	Seronegative RA	Prednisolone	Clinical improvement
			76	M	Sinovac	Seronegative RA	Prednisolone	Clinical improvement
5	An et al. [9]	China	23	F	CoronaVac	Reactive arthritis	IA Betamethasone	Remission
6	Hyun et al. [10]	South Korea	68	F	OxfordAstraZen	PMR	NSAID, and possiblly Prednisolone (it was not mentioned who received these)	Remission
			67	F	OxfordAstraZen	PMR		Remission
			67	F	OxfordAstraZen	PMR		Remission
			25	F	OxfordAstraZen	PMR		Remission
			70	F	OxfordAstrazen	PMR		Remission
7	Gentiloni et al. [11]	Italy	71	F	PfizerBioNTech	Seronegative RA	Prednisolone	Remission
			72	M	PfizerBioNTech	Seronegative RA	Prednisolone	Remission
			61	M	PfizerBioNtech	Seronegative RA	Prednisolone	Remission
			68	M	PfizerBioNTech	Seronegative RA	Prednisolone	Remission
			72	M	PfizerBioNTech	PMR	Prednisolone	Remission
			38	F	PfizerBioNTech	Undifferentiated CTD	Prednisolone	Remission
			46	F	PfizerBioNTech	DM	Prednisolone	Remission
			78	F	PfizerBioNTech	Cutaneous vasculitis	Prednisolone	Remission
8	Santiago et al. [18]	n/a	32	M	Pfizer BioNTech	Sarcoidosis	Prednisolone and azathioprine	Clinical improvement
9	Nune et al. [19]	United Kingdom	24	M	Pfizer BioNTech	SLE	Prednisolone and methotrexate	Clinical improvement
10	Kreuter et al. [20]	Germany	79	M	Pfizer BioNTech	SLE	HCQ and prednisolone	Remission
11	Zavala-Miranda et al. [21]	Mexico	23	F	Oxford-AstraZeneca	SLE	Mycopheolate, HCQ, glucocorticosteroids,	Clinical improvement
12	Hidaka et al. [22]	Japan	53	F	Pfizer BioNTech	SLE	Prednisolone	Clinical improvement
13	Raviv et al. [23]	Israel	24	M	Pfizer BioNTech	SLE	Hydroxychloroquine, topical steroids and NSAID	Clinical improvement
14	Zengarini et al. [24]	Italy	30	F	Pfizer BioNTech	SLE	Prednisolone	Clinical improvement
15	Matsuo et al. [25]	Japan	34	F	Pfizer BioNTech	Sarcoidosis	Prednisolone	Remission
16	Rademacher et al. [26]	Germany	21	F	Oxford-AstraZeneca	Sarcoidosis	Prednisolone	Clinical improvement
			27	M	Oxford-AstraZeneca	Sarcoidosis	Prednisolone	Clinical improvement
17	Kaur et al. [12]	United States	54	M	PfizerBioNTech	SLE	Prednisolone	Remission
18	Molina-Rios et al. [13]	Colombia	42	F	PfizerBioNTech	SLE	Prednisolone, HCQ and azathioprine	Remission
19	Arshadi Mousa et al. [14]	Saudi Arabia	22	F	PfizerBioNTech	SLE	Prednisolone, HCQ and azathioprine	Remission
20	Báez-Negrón et al. [15]	Puerto Rico	27	F	Moderna	SLE	Prednisolone, HCQ and MMF	No improvement
21	Patil and Patil. [16]	India	22	F	Covishield	SLE	Prednisolone, HCQ and MMF	Clinical improvement
22	Lemoine et al. [17]	United States	68	F	Pfizer BioNTech	SLE	Methotrexate and prednisone	Clinical improvement
23	Izuka et al. [27]	Japan	70	M	Moderna	PMR	Acetaminophen, was self-limited before commencing steroids	Clinical improvement
24	Ottaviani et al. [28]	France	74	F	PfizerBioNTech	PMR	Glucocorticoids	Clinical improvement
			70	F	PfizerBioNTech	PMR	Glucocorticoids and methotrexate	Clinical improvement
			74	F	PfizerBioNTech	PMR	Glucocorticoids	Clinical improvement
			77	F	PfizerBioNTech	PMR	Glucocorticoids and methotrexate	Clinical improvement
			65	M	Moderna	PMR	Glucocorticoids	Clinical improvement
			78	F	PfizerBioNTech	PMR	Glucocorticoids	Clinical improvement
			73	F	PfizerBioNTech	PMR	Glucocorticoids	Clinical improvement
			75	F	PfizerBioNTech	PMR	Glucocorticoids and tocilizumab	Clinical improvement
			77	M	PfizerBioNTech	PMR	Shoulder corticosteroid injections	Clinical improvement
			89	M	PfizerBioNTech	PMR	Glucocorticoids and methotrexate	Clinical improvement
25	Manzo et al. [30]	Italy	69	F	Pfizer BioNTech	PMR	Prednisolone	Clinical improvement
26	Gambichler et al. [31]	Germany	82	M	Pfizer BioNTech	GCA	Not reported	Not Reported
27	Sauret et al. [32]	France	70	M	Oxford-AstraZeneca	GCA	Prednisolone	Remission
28	Mejren et al. [33]	Spain	62	F	Pfizer BioNTech	GCA	Prednisolone	Clinical improvement
29	Anzola et al. [34]	Spain	83	F	Pfizer BioNTech	GCA	Pulse steroids and methotrexate	Remission
30	Lee et al. [35]	United States	34	M	Pfizer BioNTech	GCA	IV methyl prednisolone, and oral prednisolone	Clinical improvement
31	Fillon et al. [36]	France	73	M	Pfizer BioNTech	PAN	Cyclophosphamide and steroids	No improvement
32	Gillion et al. [37]	Belgium	77	M	Oxford-AstraZeneca	ANCA-negative vasculitis	Methylprednisolone	Remission
33	Feghali et al. [38]	United States	58	M	Moderna	AAV	Prednisolone, cyclophosphamide and rituximab	Remission
34	Nakatani et al. [39]	Japan	80	M	Pfizer BioNTech	LVV	Not reported	Not reported
35	Schierz et al. [40]	Germany	78	F	Moderna	LVV	Not reported	Not reported
36	Shakoor et al. [41]	United States	78	F	Pfizer BioNTech	AAV	Rituximab and prednisone	Clinical improvement
37	Hakroush et al. [42]	Germany	79	F	Pfizer BioNTech	AAV	Prednisone and cyclophosphamide	Clinical improvement
38	Baier et al. [43]	Germany	57	F	Pfizer BioNTech	AAV	Methylprednisolone and prednisone	Clinical improvement
39	Okuda et al. [44]	Japan	37	F	Pfizer BioNTech	AAV	Prednisolone	Clinical improvement
40	Obata et al. [45]	Japan	84	M	Pfizer BioNTech	AAV	Methylprednisolone and prednisolone	Clinical Improvement
41	Shirai et al. [46]	Japan	63	F	Pfizer BioNTech	AAV	Rituximab, prednisolone and cyclophosphamide	Clinical improvement
42	Al-Yafeai et al. [47]	United States	62	F	Pfizer BioNTech	AAV	Rituximab, cyclophosphamide and plasmapheresis	Clinical improvement
43	So et al. [48]	Korea	42	M	Pfizer BioNTech	AAV	Methylprednisolone, oral prednisolone, rituximab and plasma exchange	Clinical improvement
44	Al-Allaf et al. [49]	Qatar	46	M	Pfizer BioNTech	AAV	Azathioprine and prednisolone	Clinical improvement
45	Nappi et al. [50]	Italy	63	M	Moderna	AAV	Methylprednisolone, prednisolone and cyclophosamide	Clinical improvement
46	Sekar et al. [51]	USA	52	M	Moderna	AAV	Rituximab, cyclophosphamide and prednisone	No improvement
47	Ibrahim et al. [53]	USA	79	F	Moderna	AAV	Prednisone and azathioprine	Clinical improvement
48	Prabhahar et al. [52]	India	51	M	Oxford-AstraZeneca	AAV	Prednisolone and rituximab	Remission
49	Su et al. [54]	Taiwan	52	F	Oxford-AstraZeneca	Cutaneous PAN	Prednisolone and methotrexate	Remission
50	Anderegg et al. [55]	Switzerland	39	M	Moderna	ANCA negative vasculitis	Glucocorticoids and cyclophosphamide	No improvement
			81	M	Moderna	AAV	Glucocorticoids, cyclophosphamide and plasmapheresis	Remission
51	Yokote et al. [94]	Japan	71	F	Pfizer BioNTech	PMR	Prednisolone	Clinical improvement
52	Nune et al. [29]	United Kingdom	70	F	PfizerBioNTech	Seronegative RA	Prednisolone	Clinical improvement
			44	F	OxfordAstrazenic	PMR	Prednisolone	Clinical improvement
53	Vutipongsatorn et al. [56]	United Kingdom	55	F	Pfizer BioNTech	DM	IV methyl prednisolone, IVIG, cyclophosphamide and MMF	Clinical improvement
			72	F	Pfizer BioNTech	PM	IV methyl prednisolone and IVIG	Clinical improvement
54	Maramattom et al. [57]	India	74	M	OxfordAstraZen	PM	Prednisolone	Remission
			75	F	OxfordAstraZen	PM	Prednisolone and mycophenolate	Remission
			80	F	OxfordAstraZen	PM	Prednisolone	Remission
55	Coronel et al. [58]	Mexico	76	F	Pfizer BioNTech	DM	Corticosteroids and methotrexate	Clinical improvement
56	Gouda et al. [59]	Egypt	43	F	Pfizer BioNTech	DM	Prednisolone, MMF and HCQ	Clinical improvement
57	Capassoni et al. [61]	Italy	37	F	Oxford-AstraZeneca	PM	IV methyl prednisone	Clinical improvement
58	Tagini et al. [60]	Switzerland	20	F	Moderna	Behcet’s	Colchicine, prednisone and azathioprine	Clinical improvement
59	Huang et al. [62]	Taiwan	44	M	Oxford-AstraZeneca	DM	IV methylprednisolone and cyclophosphamide	ICU hospitalization and death
60	Theodorou et al. [63]	Greece	56	F	mRNA	Focal myositis	Cryotherapy, compression, and NSAIDs	Remission
61	Ramalingam et al. [64]	Mexico	81	M	Moderna	Focal myositis	Methylprednisolone	Clinical improvement
62	Gonzalez et al. [65]	United States	45	M	Moderna	DM	Methylprednisolone, rituximab, IVIG, and methotrexate	Remission
			58	F	Covishield	DM	MMF, HCQ, cyclophosphamide, rituximab, tofacitinib, tacrolimus, and plasma exchange	Clinical improvement
			45	F	PfizerBioNTech	DM	Corticosteroids, HCQ, and MMF	Clinical improvement
			28	F	PfizerBioNTech	DM	Prednisolone and cyclophosphamide	Clinical improvement
			51	F	PfizerBioNTech	DM	Corticosteroids, rituximab, tacrolimus and IVIG	Clinical improvement
			54	F	PfizerBioNTech	DM	Azathioprine which was changed to MMF due to intolerance and prednisolone	Clinical improvement
63	Yoshida et al. [66]	Japan	81	F	PfizerBioNTech	DM	Prednisolone	Clinical improvement
			87	F	PfizerBioNTech	DM	Prednisolone and IVIG	Clinical improvement
64	Gupta et al. [67]	India	46	F	Oxford-AstraZeneca	Anti-synthetase syndrome	Prednisolone and methotrexate	Clinical improvement
65	Hashizume et al. [68]	Japan	29	F	PfizerBioNTech	Behcets	Colchicine	Clinical improvement
			59	M	PfizerBioNTech	Behcets	Corticosteroid and Colchicine	Clinical improvement
66	Lebowitz et al. [86]	United States	49	M	Pfizer BioNTech	Reactive arthritis	Prednisolone	Clinical improvement
67	Baimukhamedov et al. [69]	Kazakhstan	58	M	SPUTNIK-V	Reactive arthritis	IA corticosteroid and NSAIDs	No improvement
68	Cole et al. [70]	United Kingdom	70	M	Oxford-AstraZeneca	Systemic sclerosis	Not recorded	Not reported
69	Oniszczuk et al. [71]	France	34	F	Pfizer BioNTech	Systemic sclerosis	Antihypertensives and ACE inhibitors	Clinical improvement
70	Metin et al. [72]	Turkey	55	F	Pfizer BioNTech	Localised scleroderma	Clobetasol pomade and calcipotriol pomade	Remission
71	Wireko et al. [73]	United States	69	M	Pfizer BioNTech	Pseudogout and septic arthritis	Ceftriaxone	Clinical improvement
72	Magliulo et al. [87]	United States	45	F	Moderna (2)	AOSD	Prednisolone	Remission
73	Leone et al. [88]	Italy	36	M	Oxford-AstraZeneca	AOSD	Methylprednisolone and anakinra	Clinical improvement
74	AlQudari et al. [89]	Saudi Arabia	29	M	Oxford-AstraZeneca	AOSD	Methylprednisolone	Clinical improvement
75	Park et al. [74]	Korea	36	F	Pfizer BioNTech	AOSD	Methylprednisolone and tocilizumab	Clinical improvement
76	Sweeney et al. [90]	Australia	53	M	Oxford-AstraZeneca	AOSD	Prednisolone	Remission
77	Muench et al. [91]	Germany	26	F	Pfizer BioNTech	AOSD	Methylprednisolone, IVIG, and anakinra	Clinical improvement
78	Bindoli et al. [92]	Italy	65	M	OxfordAstraZeneca	AOSD	Anakira and prednisone	Clinical improvement
			57	F	PfizerBioNTech	AOSD	Anakinra and prednisolone	Clinical improvement
			53	F	PfizerBioNTech	AOSD	Dexamethasone and anakinra	Clinical improvement
			50	F	PfizerBioNTech	AOSD	Prednisolone, anakinra, and cyclosporine	Clinical improvement
79	Iwata et al. [76]	Japan	53	F	Pfizer BioNTech	Cutaneous vasculitis	Betamethasone	Remission
80	Azzazi et al. [77]	n/a	57	F	Sinopharm	Cutaneous vasculitis	Oral prednisolone	Clinical improvement
81	Fritzen et al. [78]	Brazil	60	F	Oxford-AstraZeneca	Cutaneous vasculitis	Oral Prednisone	Remission
82	Ungari et al. [93]	USA	64	M	Oxford-AstraZeneca	Cutaneous vasculitis	Systemic antihistamine and local steroid therapy	Remission
83	Oskay et al. [79]	Turkey	77	M	CoronaVac	Cutaneous vasculitis	Prednisolone	Remission
84	Mucke et al. [80]	Germany	76	M	Pfizer BioNTech	Cutaneous vasculitis	Prednisolone	Remission
85	Altun et al. [81]	Turkey	38	M	Pfizer BioNTech	Cutaneous vasculitis	Prednisolone	Clinical improvement
86	Uh et al. [82]	Korea	64	F	OxfordAstraZeneca	Cutaneous Vasculitis	Antihistamines and topical steroids	Remission
			44	F	OxfordAstraZeneca	Cutaneous Vasculitis	Methylprednisolone, antihistamine	Remission
			68	F	OxfordAstraZeneca	Cutaneous Vasculitis	Methylprednisolone, antihistamine, topical steroid	Remission
			67	F	OxfordAstraZeneca	Cutaneous Vasculitis	Methylprednisolone, antihistamine, topical steroid	Remission
			59	F	OxfordAstraZeneca	Cutaneous Vasculitis	Methylprednisolone, antihistamine, and topical steroid	Clinical improvement
87	Fiorillo et al. [83]	Italy	71	F	Oxford-AstraZeneca	Cutaneous vasculitis	Prednisone	Remission
88	Abdelmaksoud et al. [84]	Italy	17	F	PfizerBioNTech	Cutaneous IgA vasculitis	Systemic corticosteroids	Remission
			48	M	PfizerBioNTech	Cutaneous IgA vasculitis	Systemic corticosteroids	Remission
89	Erler et al. [85]	Germany	42	F	Pfizer BioNTech	Cutaneous vasculitis	Prednisolone	Remission
90	Zhou et al. [95]	Hong Kong	72	F	Pfizer BioNTech	AOSD with myocarditis and heart failure	Prednisolone, indomethacin, hydrocortisone and methotrexate	Remission
91	Yoshino et al. [96]	Japan	56	M	Pfizer BioNTech	AAV with periaortitis (MPO) and GN	Methylprednisolone, cyclophosphamide, and methotrexate	Remission
92	Alalem et al. [97]	Saudi Arabia	24	M	OxfordAstraZen	Reactive arthritis (monoarthritis)	Ibuprofen, naproxen, and IA triamcinolone	Symptoms improved
93	Wojturska et al. [98]	Poland	33	M	OxfordAstraZen	Reactive arthritis (monoarthritis)	Diclofenac	Remission
			39	M	Moderna	Reactive arthritis (polyarthritis)	Celecoxib	Remission
			67	F	OxfordAstraZen	Reactive arthritis (polyarthritis)	Methylpredinosolone	Remission
94	Weng et al. [99]	Canada	51	F	OxfordAstraZen	AOSD	Prednisone and Celecoxib	Symptoms improved
95	Wang et al. [100]	Australia	47	F	Sinopharm	SCLE	Hydroxychloroquine	Not reported
96	VanDerVeer et al. [101]	USA	66	F	Pfizer BioNTech	RA	Indomethacin	Symptoms improved
			61	F	Pfizer BioNTech	RA	Prednisolone and methotrexate	Clinical remission
			36	F	Morderna	RA	Adalimumab	Clinical remission
			72	F	Pfizer BioNTech	RA	Prednisolone, leflunomide, and golimumab	Clinical remission
			69	M	Pfizer BioNTech	PMR	Prednisolone	Symptoms improved
97	Tosunoğlu et al. [102]	Taiwan	21	F	Pfizer BioNTech	PM	Methylprednisolone and IVIG	Symptoms improved
98	Sugimoto et al. [103]	Japan	62	F	Pfizer BioNTech	DM (anti-MDA-5)	Methylprednisolone, oral tacrolimus, IV cyclophosphamide, and plasmapheresis	Patient died
99	Srichawla et al. [104]	USA	59	F	Moderna	PAN	Methylprednisolone, and methotrexate	Remission
100	Sogbe et al. [105]	Spain	72	F	Pfizer BioNTech	SLE with myocarditis	prednisolone	Remission
101	Shukla et al. [106]	India	56	F	OxfordAstraZen	Sarcoidosis	prednisolone	Symptoms improved
102	Shokraee et al. [107]	Iran	41	M	Sputnik V	Reactive arthritis (monoarthritis)	IA injection of triamcinolone, and oral prednisolone	Symptoms improved
103	Sakai et al. [108]	Japan	26	F	Pfizer BioNTech	SLE	Methylprednisolone, HCQ, MMF, and belimumab	Not reported
			62	M	Morderna	SLE	Methylprednisolone, prednisolone, and IVcyclophosphamide	Symptom improved
104	Saiz et al. [109]	Spain	48	F	Moderna	AAV (PR3 positive)	corticosteroids, methotrexate, and rituximab	Symptoms improved
105	Rimmer et al. [110]	USA	79	F	Moderna	SCLE	HCQ, prednisolone, IVIG, and mycophenolate	Remission
106	Ohkubo et al. [111]	Japan	61	M	Pfizer BioNTech	PAN	Prednisolone	Remission
107	Wang et al. [112]	USA	60	F	Pfizer BioNTech	Amyopathic DM (anti-MDA-5)	Prednisolone	Not reported
108	Mung et al. [113]	UK	71	M	OxfordAstraZen	Seronegative RA	Prednisolone and HCQ	Symptoms improved
109	Lo Sardo et al. [114]	Italy	78	M	OxfordAstraZen	GCA	Prednisolone and Tocilizumab	Symptoms improved
110	Koh et al. [115]	Taiwan	17	F	Pfizer BioNTech	Seronegative RA	Oral NSAIDs, arthrocentesis, etanercept, and sulfasalazine	Symptoms improved
111	Kawamura et al. [116]	Japan	71	F	Pfizer BioNTech	AAV(MPO)	Corticosteroids and IV cyclophosphamide	Symptoms improved
112	Katsouli et al. [117]	Greece	52	F	OxfordAstraZen	LVV	Prednisolone and tocilizumab	Remission
113	Kan et al. [118]	China	72	F	Pfizer BioNTech	AOSD with myocarditis and heart failure	Prednisolone	Remission
114	Iwata et al. [119]	Japan	53	F	Pfizer BioNTech	Cutaneous vasculitis	Betamethasone	Remission
115	Haruna et al. [120]	Japan	77	F	Pfizer BioNTech	PMR without shoulder symptoms	Prednisolone	Symptoms improved
116	Golstein et al. [121]	Belgium	48	F	Pfizer-BioNTech	Reactive arthritis (oligoarticular)	Prednisolone	symptom improved
			61	M	Pfizer-BioNTech	Reactive arthritis (oligoarticular)	Prednisolone	symptom improved
			59	F	Pfizer-BioNTech	Reactive arthritis (oligoarticular)	Prednisolone	symptom improved
			53	F	Moderna	Reactive arthritis (oligoarticular)	Prednisolone	symptom improved
			57	F	Pfizer-BioNTech	Reactive arthritis (oligoarticular)	Prednisolone	symptom improved
			66	F	Oxford-Astrazeneca	Reactive arthritis (oligoarticular)	Prednisolone	symptom improved
			31	F	Pfizer-BioNTech	Reactive arthritis (oligoarticular)	Prednisolone	symptom improved
			44	F	Pfizer-BioNTech	Reactive arthritis (oligoarticular)	Prednisolone	symptom improved
			80	F	Pfizer-BioNTech	Reactive arthritis (oligoarticular)	Prednisolone	symptom improved
			68	M	Pfizer-BioNTech	Reactive arthritis (oligoarticular)	Prednisolone	symptom improved
			45	F	Pfizer-BioNTech	Reactive arthritis (oligoarticular)	Prednisolone	symptom improved
			21	F	Pfizer-BioNTech	Reactive arthritis (oligoarticular)	Prednisolone	symptom improved
			67	M	Pfizer-BioNTech	Reactive arthritis (oligoarticular)	Prednisolone	symptom improved
			55	M	Pfizer-BioNTech	Reactive arthritis (oligoarticular)	Prednisolone	symptom improved
			52	F	Pfizer-BioNTech	Reactive arthritis (oligoarticular)	Prednisolone	symptom improved
			44	F	Pfizer-BioNTech	Reactive arthritis (oligoarticular)	Prednisolone	symptom improved
			29	F	Pfizer-BioNTech	Reactive arthritis (oligoarticular)	Prednisolone	symptom improved
117	Gen et al. [122]	Japan	82	F	Moderna	AAV (MPO)	Prednisolone	Symptoms improved
118	Furr et al. [123]	USA	69	F	Moderna	PMR	Prednisolone	Symptoms improved
			74	M	Pfizer-BioNTech	PMR	Prednisolone	Symptom improved
119	El Hasbani et al. [124]	USA	47	F	Pfizer-BioNTech	AAV (MPO)	Methylprednisolone, prednisone and azathioprine	Symptoms improved
120	Durucan et al. [125]	Turkey	24	M	Pfizer-BioNTech	PM and myocarditis	Myorelaxant, and NSAIDs	Remission
121	Ansari et al. [126]	Iran	28	M	Oxford-Astrazeneca	IMNM	Methylprednisolone, prednisone, and azathioprine.	Remission
122	Albers et al. [127]	Germany	41	M	Pfizer-BioNTech	Sarcoidosis	topical corticosteroids, tacrolimus, and HCQ	Symptoms improved
123	Alalem et al. [97]	Saudi Arabia	24	M	Oxford-Astrazeneca	Reactive arthritis (monoarthritis)	Ibuprofen, Naproxen, and IA triamcinolone	Symptoms improved
124	Zamoner et al. [128]	Brazil	58	F	Oxford-Astrazeneca	AAV (MPO) with crescentic GN	Methylprednisolone, prednisone, IV cyclophosphamide, and azathioprine	Not reported
125	Yonezawa et al. [129]	Japan	54	M	Pfizer-BioNTech	RA	Methylprednisolone, and iguratimod	Symptoms improved
126	Yadav et al. [130]	Nepal	52	F	Johnson and Johnson	AAV (C-ANCA positive) with rapidly progressing GN	Cyclophophaide and methylprednisolone	Not reported
127	Xia et al. [131]	Australia	68	M	Oxford-Astrazeneca	GCA(AION)	Methylprednisolone and tocilizumab	Not reported
128	Wu et al. [132]	USA	77	F	Pfizer-BioNTech	DM (anti-TIF positive)	IV methylprednisolone, IVIG, MMF, and prednisolone	Symptoms improved
129	Watanabe et al. [133]	Japan	53	M	Pfizer-BioNTech	Seropositive RA	Prednisolone, methotrexate, and tocilizumab	Remission
130	Wang et al. [134]	Taiwan	81	M	Oxford-Astrazeneca	SCLE (positive Ro, ANA)	Prednisolone	Symptoms improved
131	Vanaskova et al. [135]	Czech Republic	53	M	Pfizer-BioNTech	Reactive arthritis (monoarthritis)	Dexamethasone	Symptoms improved
132	Uddin et al. [136]	Pakistan	59	M	Pfizer-BioNTech	AAV(PR3)	Methylprednisolone, rituximab, and prednisone	Symptoms improved
133	Suzuki et al. [137]	Japan	72	M	Pfizer-BioNTech	AAV (MPO)	Methylprednisolone, prednisolone, and rituximab	Symptoms improved
134	Seeley et al. [138]	USA	35	F	Pfizer-BioNTech	APL syndrome (catastrophic)	IV dexamethasone, HCQ, and prednisolone	Symptoms improved
135	Schoenardie et al. [139]	Brazil	25	F	Oxford-Astrazeneca	Seronegative RA (monoarthritis)	NSAID and prednisolone	Symptoms improved
136	Park et al. [140]	USA	64	M	Moderna	Seronegative RA (monoarthritis)	Naproxen	Symptoms improved
			73	F	Moderna	Dactylitis	Celecoxib	Symptoms improved
137	Ohmura et al. [141]	Japan	41	F	Moderna	MVV (negative ANCA)	Ibuprofen and prednisolone	Symptoms improved
138	Numakura et al. [142]	Japan	61	M	Pfizer-BioNTech	Sarcoidosis	IA steroid injection	Not reported
139	Mohaghegh et al. [143]	Iran	65	F	COVIran Barekat	Sarcoidosis	Methotrexate and prednisolone	Symptoms improved
140	Magen et al. [144]	Israel	34	F	Pfizer-BioNTech	PM	IV methylprednisolone, prednisone, IVIG, and azathioprine	Symptoms persists
141	Ma et al. [145]	China	70	F	CoronaVac	AAV(MPO) GN	IV Glucocorticoids, IV cyclophosphamide, and low-dose steroids maintenance therapy	Symptoms improved
142	Lourenço et al. [146]	Portugal	75	F	Oxford-Astrazeneca	PMR	IM betamethasone and prednisolone	Symptoms improved
143	Kreuter et al. [147]	Germany	68	F	Pfizer-BioNTech	IIM	IV glucocorticosteroids	Symptoms improved
144	Kim, Y et al. [148]	South Korea	77	F	Pfizer-BioNTech	SVV with crescentic GN	Methylprednisolone	Remission
145	Kim, J et al. [149]	South Korea	30	M	Pfizer-BioNTech	DM	Glucocorticoid and azathioprine	Not reported
146	Kim B, et al. [150]	South Korea	72	F	Moderna	AAV (MPO +ve) with crescentic GN	Plasmapheresis, IV cyclophosphamide and IV methylprednisolone	Constitutional symptoms improved
147	Khanna et al. [151]	USA	18	F	Pfizer-BioNTech	SLE, with neutrophilic urticarial dermatosis	Prednisone and HCQ	Remission
148	Holzer et al. [152]	Germany	19	M	Pfizer-BioNTech	DM	Glucocorticoids, IVIG, Tofacitinib, MMF, Rituximab, Ciclosporin A, Anakinra, Nintedanib and Daratumumab	Not reported
			57	F	Pfizer-BioNTech	DM	Glucocorticoids, HCQ, and Azathioprine	Not reported
			51	M	Pfizer-BioNTech	DM	Glucocorticoids, SC Methotrexate, HCQ, and Azathioprine	Not reported
149	Greb et al. [153]	USA	79	M	Pfizer-BioNTech	GCA	Prednisolone	Remission
150	Godoy et al. [154]	Brazil	64	F	Oxford-Astrazeneca	PM	Corticoid and immunosuppressant therapy	Symptoms improved
151	Gamonal et al. [155]	Brazil	27	F	Oxford-Astrazeneca	SLE, with alopecia areata	Prednisolone and hydroxychloroquine	Not reported
152	Farooq et al. [156]	UK	63	M	Oxford-Astrazeneca	Inflammatory myositis with myocarditis and pneumonitis	Methylprednisolone and prednisolone	Symptoms improved
153	Essien et al. [157]	USA	27	M	Pfizer-BioNTech	AAV	Prednisolone and rituximab	Relapse
154	Draz et al. [158]	Egypt	54	M	Pfizer-BioNTech	Reactive arthritis	IM dexamethasone and NSAID	Symptoms improved
155	Chua et al. [159]	Taiwan	30	F	Moderna	AOSD	methylprednisolone, prednisolone, and naproxen	Remission
156	Christodoulou et al. [160]	Greece	72	F	Moderna	AAV (MPO) with pulmonary-renal syndrome	Prednisolone, cyclophosphamide and plasmapheresis	Remission
157	Chomičienė et al. [161]	Lithuania	49	F	Pfizer-BioNTech	HSP (IgA vasculitis)	Methylprednisolone and plasmapheresis	Remission
			65	F	Pfizer-BioNTech	Urticarial vasculitis	IVdexamethasone, plasmapheresis and oral methylprednisolone	Remission
158	Chaima et al. [162]	Tunisia	52	F	Pfizer-BioNTech	DM	Prednisolone	Remission
159	Bose et al. [163]	India	53	M	Oxford-Astrazeneca	Focal myositis	NSAIDa	Symptoms improved
160	Barman et al. [164]	India	38	F	Oxford-Astrazeneca	RA	prednisolone, methotrexate, HCQ, and sulfasalazine	Symptom improved
161	Ball-Burack et al. [165]	USA	22	M	Johnson and Johnson	LCC vasculitis	NSAIDs	Symptoms improved
162	Aoki et al. [166]	Japan	81	M	Pfizer-BioNTech	LVV	Naproxen	Symptoms improved
163	Ahmer et al. [167]	Australia	50	F	Oxford-Astrazeneca	Cryoglobulinemic vasculitis	Nil	Remission
164	Villa et al. [168]	Spain	63	M	Moderna	AAV (MPO) with crescentic GN	IV glucocorticoids, prednisone, and cyclophosphamide	Symptoms improved
165	Türk et al. [169]	Turkey	72	F	Sinovac	Reactive arthritis	Prednisolone	Symptoms improved
			79	F	Sinovac	Reactive arthritis	Methylprednisolone	Symptoms persisted
166	Sirufo et al. [170]	Italy	76	F	Oxford-Astrazeneca	HSP	Deflazacort	Symptoms improved
167	Risal et al. [171]	Nepal	47	F	Oxford-Astrazeneca	AOSD	Prednisolone and methotrexate	Symptoms improved
168	Quattrini et al. [172]	Italy	83	F	Pfizer-BioNTech	RS3PE	Prednisolone and methotrexate	Rapid clinical improvement
169	Naitlho et al. [173]	Morroco	62	M	Oxford-Astrazeneca	HSP	Prednisolone	Symptoms improved
170	Liu et al. [174]	USA	70	M	Pfizer-BioNTech	SACL	Topical steroids	Symptoms improved
171	Kar et al. [175]	India	46	M	COVAXIN	Cutaneous small vessel vasculitis	Nil	Symptoms improved
172	Hines et al. [176]	USA	40	F	Pfizer-BioNTech	HSP	Nil	Symptoms improved
173	Guzmán-Pérez et al. [177]	Spain	57	F	Oxford-Astrazeneca	Cutaneous small-vessel vasculitis	Nil	Not reported
174	Chen et al. [178]	Taiwan	70	F	Moderna	AAV (MPO) with pulmonary-renal syndrome	Plasma exchange, corticosteroid, and anti-CD20 therapy	Symptoms persisted
175	Chan-Chung et al. [179]	Singapore	62	F	Pfizer-BioNTech	EGPA (MPO-positive)	IV methylprednisolone and rituximab	Symptoms improved
176	Chan et al. [180]	Australia	79	F	Oxford-Astrazeneca	Polyarthralgia	Nil	Symptoms improved
177	Berry et al. [181]	USA	65	M	Janssen	Cutaneous small-vessel vasculitis	Prednisolone	Remission
178	Dube et al. [182]	USA	29	F	Pfizer-BioNTech	AAV (PR3 positive) with GN	Methylprednisolone, rituximab, and cyclophosphamide	Not reported
179	Takenaka et al. [183]	Japan	75	F	Pfizer-BioNTech	AAV (MPO positive) optic perineuritis	Methylprednisolone	Not reported
180	Tasnim et al. [184]	USA	71	M	Pfizer-BioNTech	IgG4 disease	Nil	Not reported
181	Aochi et al. [185]	Japan	78	F	Pfizer-BioNTech	IgG4 disease	Prednisolone	Not reported
			65	M	Pfizer-BioNTech	IgG4 disease	Prednisolone	Not reported
			63	M	Pfizer-BioNTech	IgG4 disease	Prednisolone	Not reported
182	Matsuda et al. [186]	Japan	59	F	Pfizer-BioNTech	AOSD	Corticosteroid and tocilizumab	Not reported
			77	F	Pfizer-BioNTech	AOSD	IV methylprednisolone, prednisolone, and tocilizumab	Not reported
			35	M	Moderna	AOSD	prednisolone	Not reported
183	Avalos et al. [187]	USA	74	F	Pfizer-BioNTech	Microscopic polyangiitis	Methylprednisolone and rituximab	Not reported
184	Sagy et al. [188]	Israel	24	M	Pfizer-BioNTech	SLE	HCQ, topical steroid, etoricoxib	Not reported
			24	M	Pfizer-BioNTech	SLE	HCQ, prednisolone, azathioprine	Not reported
			56	M	Pfizer-BioNTech	SLE	HCQ and etoricoxib	Not reported
185	Roy et al. [189]	India	60	F	Covishield	HSP	Nil	Not reported
186	Chan et al. [190]	Canada	53	F	Oxford-Astrazeneca	DM (PL12- positive) with ILD and Myocarditis	Prednisolone, MMF, methotrexate, and HCQ	Symptoms improved
			76	F	Pfizer-BioNTech	DM (SAE-1 positive) with ILD and Myocarditis	Prednisolone, MMF, and IVIG	Symptoms improved
187	Nahra et al. [191]	USA	71	M	Pfizer-BioNTech	Seronegative RA	NSAID and prednisolone	Symptoms improved
			74	M	Pfizer-BioNTech	Seropositive RA	Prednisolone and leflunomide	Symptoms improved
188	Bansal et al. [192]	USA	40	F	Pfizer-BioNTech	Seropositive RA	Methotrexate and HCQ	Not reported
189	Parperis et al. [193]	Greece	80	M	Pfizer-BioNTech	RS3PE	Prednisolone	Symptoms improved
190	Baimukhamedov et al. [194]	Kazakhstan	38	F	SPUTNIK-V	Seropositive RA	NSAIDs, methylprednisolone, and methotrexate	Not reported

IA: intra articular; IM: intramuscular; IV: intravenous; SC: subcutaneous; RA: rheumatoid arthritis; IIM: idiopathic inflammatory myositis; PM: polymyositis; DM: dermatomyositis; GCA: giant cell arteritis; NSAID: non-steroidal anti-inflammatory drugs; DMARDs: diseases modifying anti-rheumatic drugs; HCQ: hydroxychloroquine; MMF: mycophenolate mofetil; IVIG: intravenous immunoglobulin; AOSD: adult-onset stills disease; PAN: polyarteritis nodosa; SLE: systemic lupus erythematosus; RS3PE: remitting seronegative symmetrical synovitis with pitting oedema; ILD: interstitial lung disease; HSP: Henoch-Schoenlein purpura; GN: glomerulonephritis; EGPA: eosinophilic granulomatosis with polyangiitis; SACL: subacute cutaneous lupus; AAV: ANCA-associated vasculitis; LVV: large-vessel vasculitis; MVV: medium-vessel vasculitis; SVV: small-vessel vasculitis; LCC: leukocytoclastic; APL: Antiphospholipid; AION: anterior ischemic optic neuropathy; HCQ: hydroxychloroquine; IMNM: immune mediated necrotizing myopathy; CTD: connective tissue disease.

**Table 2 vaccines-11-01571-t002:** Demographics of patients included in the systematic review, presenting with new-onset R-IMID post-COVID-19 vaccination.

Variable	Number (N = 271)
Age	56 mean (±18.9 SD)
Gender	
Male	101 (37.1%)
Female	170 (62.5%)
Country	
United States	47 (17.3%)
Japan	36 (13.3%)
Italy	23 (8.5%)
Belgium	18 (6.6%)
Germany	15 (5.5%)
South Korea	15 (5.5%)
India	14 (5.1%)
France	13 (4.8%)
United Kingdom	8 (2.9%)
Turkey	8 (2.9%)
Taiwan	7 (2.6%)
Spain	6 (2.2%)
Israel	5 (1.8%)
Brazil	5 (1.8%)
Australia	5 (1.8%)
Greece	4 (1.5%)
Mexico	3 (1.1%)
Canada	3 (1.1%)
Switzerland	3 (1.1%)
Poland	3 (1.1%)
Saudi Arabia	3 (1.1%)
Iran	3 (1.1%)
China	3 (1.1%)
Moraco	1(0.4%)
Kazakhstan	2 (0.7%)
Egypt	2 (0.7%)
Nepal	2 (0.7%)
Lithuania	2 (0.7%)
Portugal	1 (0.4%)
Singapore	1 (0.4%)
Hongkong	1 (0.4%)
Qatar	1 (0.4%)
Puerto Rico	1 (0.4%)
Pakistan	1 (0.4%)
Columbia	1 (0.4%)
Tunisia	1 (0.4%)
Czech Republic	1 (0.4%)
Unknown	3 (1.1%)

R-IMID: rheumatic immune-mediated inflammatory disease; SD: standard deviation.

**Table 3 vaccines-11-01571-t003:** Vaccination characteristics of patients with new-onset R-IMID post-COVID-19 vaccinations.

Variable	Number (N = 271)
Vaccination types	
Pfizer BioNTech	153 (56.5%)
Oxford-AstraZeneca	61 (22.5%)
Moderna	33 (12.2%)
Corona Vac/Sinovac	07 (2.6%)
Covishield	03 (1.1%)
Sputnik-V	03 (1.1%)
Johnson and Johnson	02 (0.7%)
Sinopharm	02 (0.7%)
Covaxin	01 (0.4%)
Janssen	01 (0.4%)
COVIran Barekat	01 (0.4%)
Information not available	04 (1.5%)
Number of doses	
One	119 (43.9%)
Two	123 (45.4%)
Three	11 (4.1%)
Not reported	18 (6.6%)

**Table 4 vaccines-11-01571-t004:** Summary of the distribution of the new-onset R-IMID following SARS-CoV-2 vaccinations.

R-IMID	Number (N = 271)	%
Vasculitides	86	31.7%
Small-vessel vasculitis	64	23.6%
(1) ANCA-associated vasculitis	30	11.1%
(a) Eosinophilic granulomatosis with polyangiitis (EGPA)	3	1.1%
(2) ANCA-negative vasculitis	3	1.1%
(3) Cutaneous vasculitis	22	8.1%
(4) Henoch-Schonlein purpura (HSP)	5	1.8%
(5) Cryoglubulinemia	1	0.4%
Medium-vessel vasculitis	6	2.2%
Large-vessel vasculitis	8	3.0%
1. IGG4 disease	4	1.5%
Giant-cell arteritis	8	3.0%
Connective tissue diseases	66	24.4%
1. Idiopathic inflammatory myositis (IIM)	37	13.7%
(a) Polymyositis (PM)	9	3.3%
(b) Dermatomyositis (DM)	24	8.9%
(c) Focal myositis	3	1.1%
(d) Immune-mediated necrotising myopathy (IMNM)	1	0.4%
2. Antisynthetase syndrome	1	0.4%
3. Systemic lupus erythematosus (SLE)	24	8.9%
(a) Subacute cutaneous lupus (SCLE)	6	2.2%
4. Antiphospholipid syndrome	1	0.4%
5. Undifferentiated CTD	1	0.4%
6. Systemic sclerosis	2	0.7%
Inflammatory arthritis	55	20.3%
1. Reactive arthritis	30	11.1%
2. Rheumatoid arthritis	21	7.7%
(a) Seropositive	9	3.3%
(b) Seronegative	12	4.4%
(i) Monoarthritis	2	0.7%
3. Crystal arthritis (pseudogout)	1	0.4%
4. Remitting seronegative symmetrical synovitis with pitting oedema (RS3PE)	2	0.7%
5. Dactylitis	1	0.4%
Polymyalgia rheumatica	21	7.7%
Adult-onset stills disease	22	8.1%
Bechet’s disease	3	1.1%
Sarcoidosis	8	3.0%
Miscellaneous	10	3.7%
(1) Polyarthralgia and myalgia	6	2.2%
(2) Tenosynovitis	2	0.7%
(3) Localised scleroderma	1	0.4%
(4) Inflammatory myositis with myocarditis and pneumonitis	1	0.4%

R-IMID: rheumatic immune-mediated inflammatory disease; ANCA: antineutrophilic cytoplasmic antibody; CTD: connective tissue disease.

## Data Availability

The authors have primary data and agree to allow the journal to review the data if requested. Appendix A are available at the Vaccine Journal online.

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
