# Peer review of "New-Onset Rheumatic Immune-Mediated Inflammatory Diseases Following SARS-CoV-2 Vaccinations until May 2023: A Systematic Review"

_vaccines, 2023, doi:10.3390/vaccines11101571_

Round 1

Reviewer 1 Report

Dear Editor, thank you very much for the opportunity to read and evaluate this very interesting manuscript (vaccines-2633317-peer-review-v1), and indeed the scientific society needs it. It collected, summarized, tableted, analyzed, and systematically reviewed all new-onset Rheumatic Immune-Mediated Inflammatory Diseases following SARS-CoV-2 Vaccinations until May 2023 data, it is appreciated efforts. The anti-vaccination group as well as those who are hesitant people may change their minds in light of this study's conclusion and positive message. Billions of COVID-19 vaccine doses have already been administered worldwide, and one of the adverse reactions (R-IMIDs) is that depicted in the current manuscript. The manuscript is very well written, designed, and organized, which candidate to accept after minor comments.

minor comments for authors:
-The authors recommend explaining why they did not use search engines such as SCOPUS and Web of Science. Or vaccine adverse reactions such as Vaccine Adverse Event Reporting System (VAERS) and other databases.

- YOU sometimes use R-IMID instead of R-IMIDS? SUCH AS LINES 89, 95 for example.

Author Response

Thank you for reviewing our manuscript. We have read your comments and agree with your suggestions. We have revised the main manuscript accordingly. Please see our responses below.

Reviewer 1:

Comment: Dear Editor, thank you very much for the opportunity to read and evaluate this very interesting manuscript (vaccines-2633317-peer-review-v1), and indeed the scientific society needs it.

It collected, summarized, tableted, analysed, and systematically reviewed all new-onset Rheumatic Immune-Mediated Inflammatory Diseases following SARS-CoV-2 Vaccinations until May 2023 data, it is appreciated efforts. The anti-vaccination group as well as those who are hesitant people may change their minds in light of this study's conclusion and positive message. Billions of COVID-19 vaccine doses have already been administered worldwide, and one of the adverse reactions (R-IMIDs) is that depicted in the current manuscript.

The manuscript is very well written, designed, and organized, which candidate to accept after minor comments.

Our response: Thank you for your kind words. We are glad to hear that you found our article worthy of publication.

minor comments for authors:
Comment: The authors recommend explaining why they did not use search engines such as SCOPUS and Web of Science. Or vaccine adverse reactions such as Vaccine Adverse Event Reporting System (VAERS) and other databases.

Our response:  

Thank you for your comment on search engines use in our systematic review. We have robust coverage from the databases we have searched, and our inclusion of Google Scholar (GS) in addition to the key databases advised by the Cochrane handbook* (Medline, Embase and Cochrane as a minimum) is likely to have reduced the chance of missing anything of potential relevance from SCOPUS or Web of Science (WOS).

*Higgins JPT, Thomas J, Chandler J, Cumpston M, Li T, Page MJ, Welch VA (editors). 

Cochrane Handbook for Systematic Reviews of Interventions version 6.4 (updated August 2023). Cochrane, 2023. Available from www.training.cochrane.org/handbook. Sources to Search (section 4.3)  https://training.cochrane.org/handbook/current/chapter-04#section-4-3-1-1 (please see the second paragraph)

A recent article (please see the reference below) from the Journal of Informetrics has analysed the coverage of GS compared to that achieved by WOS and Scopus. In their abstract, they state that “GS found nearly all the WoS (95%) and Scopus (92%) citations”.

Alberto Martín-Martín, Enrique Orduna-Malea, Mike Thelwall, Emilio Delgado López-Cózar, Google Scholar, Web of Science, and Scopus: A systematic comparison of citations in 252 subject categories,Journal of Informetrics,Volume 12, Issue 4,2018,Pages 1160-1177

https://doi.org/10.1016/j.joi.2018.09.002.(https://www.sciencedirect.com/science/article/pii/S1751157718303249

Based on the above, it is highly unlikely that our searches would have missed any key references.

VAERS is a passive reporting system that relies on individuals sending in reports of their experiences with vaccines. Anyone can report an adverse event to VAERS.  Our SR included published new-onset R-IMIDs in the literature. Indeed, the use of self-reported data collected online has been discussed as a factor causing misclassification. Please see the following article about misreporting PMR and GCA as AEFI following COVID-19 vaccination on the VigiBase database which similar to VAERS anyone can report to.

Manzo C, Castagna A. Comment on: Risk of giant cell arteritis and polymyalgia rheumatica following COVID-19 vaccination: a global pharmacovigilance study. Rheumatology (Oxford) 2022; 61(4):e101-e102. DOI: 10.1093/rheumatology/keab849.

However, we have searched Lit Covid, and due to its prominent interest in COVID-19 vaccination, publishers and database providers are still alerted to ensuring coverage of the topic across platforms.

Comment: YOU sometimes use R-IMID instead of R-IMIDS? SUCH AS LINES 89, 95 for example.

Our response:

Thank you for your query. We used R-IMID (rheumatic immune-mediated inflammatory disease) OR R-IMIDs (rheumatic immune-mediated inflammatory diseases) based on its context. For singular, we used R-IMID or for pleural R-IMIDs. 

Line 89: We used R-IMID here as we already used ‘’new-onset cases’’ before R-IMID. 

Line 95: As you suggested, we changed R-IMID to R-IMIDs here to fit the context. 

Reviewer 2 Report

In this paper the Author reported a large series of already published cases of rheumatic disease whose onset is related to SARS-CoV2 Vaccination. The topic is very interesting and it deserves to be highlight as it puts the attention on a relevant (both from and epidemiological and pathogenetic point of view) issue related to rheumatic diseases etiology.

The title is informative, the aim as well as the methods are clear. Results support the conclusions.

COMMENTS

- Is it possible to cluster data according the number of doses received? Moreover, taking into account the distribution of vaccines in each country, is it possible to figure out if Pfizer is actually the most related to the development of rheumatic disease?

-P19L35-etc. there is not the 3.7 subtitle. Please, make the subtitles organization congruent with table 4 order or viceversa

- P9L317 correct: Moderna, not Astra Zeneca

- Do you think that case reports/series about rheumatic disease onset after anti SAS-CoV2 vaccination are published rarely? Could this issue be underestimated. Please discuss this point.

Author Response

Thank you for reviewing our manuscript. We have read your comments and agree with your suggestions. We have revised the main manuscript accordingly. Please see our responses below.

Reviewer 2:

Comment: In this paper the Author reported a large series of already published cases of rheumatic disease whose onset is related to SARS-CoV2 Vaccination. The topic is very interesting and it deserves to be highlight as it puts the attention on a relevant (both from and epidemiological and pathogenetic point of view) issue related to rheumatic diseases etiology.

The title is informative, the aim as well as the methods are clear. Results support the conclusions.

Our response: Thank you for finding our work interesting, relevant, which highlights the R-IMIDs epidemiology and pathogenesis following SARS-CoV-2 vaccinations.

Comment: Is it possible to cluster data according the number of doses received?

Our response:

Thank you for your suggestion. For clarity, due to our extensive data, we presented the vaccines used and the data on the number of doses received separately in our comprehensive review. The information about the number of doses received is available in Table 3, and Table 1 has all the details about the vaccinations and their diagnoses. We hope this is agreeable.

Comment: Moreover, taking into account the distribution of vaccines in each country, is it possible to figure out if Pfizer is actually the most related to the development of rheumatic disease?

Our response:

Thank you for your comment. Although most cases of new-onset R-IMID received the Pfizer vaccination, it is difficult to say whether the Pfizer vaccine is more likely than other vaccines to cause new-onset R-IMID cases.

While some studies provide some information about an association between Pfizer vaccines and rheumatic diseases, we do not have enough data to determine a firm association through secondary data analysis. It is difficult to figure out if Pfizer vaccine is the most associated to developing rheumatic diseases unless we have data with sufficient statistical power to infer a strong correlation corroborated with good regression models. If the existing data is normalized by vaccine distribution, the association may disappear or significantly weaken.

Comment: P19L35-etc. there is not the 3.7 subtitle. Please, make the subtitles organization congruent with table 4 order or vice versa

Our response: Thank you for pointing out an error. We have now amended it so that 3.7 represents inflammatory arthritis. We also adjusted the order of the subtitles in line with the manuscript text.

Comment: P9L317 correct: Moderna, not Astra Zeneca

Our response: Thank you for noticing this error. We have now replaced AstraZeneca with Moderna.

Comment: Do you think that case reports/series about rheumatic disease onset after anti SAS-CoV2 vaccination are published rarely? Could this issue be underestimated. Please discuss this point.

Our response:

We agree that there are many causes for underreporting R-IMID cases in the literature. It is possible that many cases of new-onset R-IMIDs were not reported due to a lack of awareness and the laborious process of publishing case reports with a high rejection rate.

As we know, in scientific reporting, the absence of evidence does not necessarily imply evidence absence. So, under-reporting may exist and be driven by the novelty of the virus that has shifted research efforts primarily towards vaccine production rather than continual research monitoring of the vaccines.

It is also worth noting that rheumatic disease outbreaks following immunization are reported in a relatively small proportion of individuals compared to the large population receiving SARs-CoV-2 vaccinations.